# Predicting the ritonavir crisis by revisiting the polymorph landscape with crystal structure prediction and form 4 structure solution
Luca Iuzzolino [1] ✉, Andrew W. Kelly[2], Mohammad T. Chaudhry[2], Cristian Jandl[3], Danny Stam[3] & Alfred Y. Lee[2]

The transformation of ritonavir form 1 into a less soluble form 2 is the most notorious example of the risks associated with crystal polymorphism in pharmaceuticals. Since then, significant advancements have occurred in the field of theoretical crystal structure prediction, which forecasts the potential polymorphs of a molecule and their stability ranking. However, a question remains whether in silico modeling would have predicted the ritonavir disaster and informed appropriate action. Furthermore, the experimental landscape of ritonavir remains incomplete as no solution of form 4 has been deposited. Here, we show that CSP would have foreseen the existence of more stable then-unfound form 2 of ritonavir at room temperature. From a risk standpoint, the threat posed by this polymorph would have been considered severe due to its unique conformational and structural characteristics, combined with the formulation's low tolerance for solubility reduction. This would have prompted additional work that could have averted the crisis. Furthermore, we determined the crystal structure of form 4 of ritonavir by three-dimensional electron diffraction, combined with in silico modeling and experimental powder X-ray diffraction, revealing a disordered motif and proving it is thermodynamically unstable.

Ritonavir (Fig. 1) needs no introduction in the solid-state community[1–11]. The polymorphic transformation of this marketed human immunodeficiency virus (HIV) drug from form 1 to the more stable and less soluble form 2 caused huge reputational and financial losses to Abbott Laboratories and left patients untreated during the reformulation process[6,11]. Often referred to as a "catastrophe", a "crisis", and a "disaster" in both scientific and generalist literatures[8,12–15], it is regarded as a milestone in raising awareness of the risks that crystal polymorphism poses to the development of safe and effective drugs[16,17]. Therefore, this event is viewed as a catalyst that prompted the pharmaceutical industry to adopt more thorough and comprehensive solid form screening practices, while also directing regulatory authorities to the application of increased scrutiny in regards to polymorph-related risks[18,19]. One key development has been the integration of experimental screening work with computational crystal structure prediction (CSP) methods[8,20–26]. Indeed, what happened to ritonavir is often mentioned as the rationale for performing CSP for all key small-molecule drug candidates, even when the experimental solid form landscape looks clear: something unexpected can always be lurking in the shadows[5,27].

Several aspects of the ritonavir polymorphism story remain unresolved. A complete CSP study of ritonavir—clearly showing the severity of the threat posed by form 2—has, to the best of our knowledge, yet to be reported in the literature. Although CSP was not fully developed in the 1990s[28], it remains of great interest to study if a risk assessment based on current computational approaches would have helped prevent the crisis. There are difficulties: ritonavir's large size and high conformational flexibility increase the computational cost and the complexity of CSP calculations. While ritonavir has often been considered beyond the capabilities of the most accurate CSP algorithms, recent progress, including the emergence of faster compute nodes, the increasing availability of Cloud computing[29], the improvement in force fields allowing a reduction in the number of required quantum-mechanical calculations[30], and the development of more accurate, computationally feasible methods for free energy calculation at

[1]Modeling & Informatics, Discovery Chemistry, Merck & Co. Inc., Rahway, NJ, USA. [2]Analytical Research & Development, Merck & Co. Inc., Rahway, NJ, USA. [3]ELDICO Scientific AG, Switzerland Innovation Park Basel Area, Allschwil, Switzerland. ✉e-mail: luca.iuzzolino@merck.com

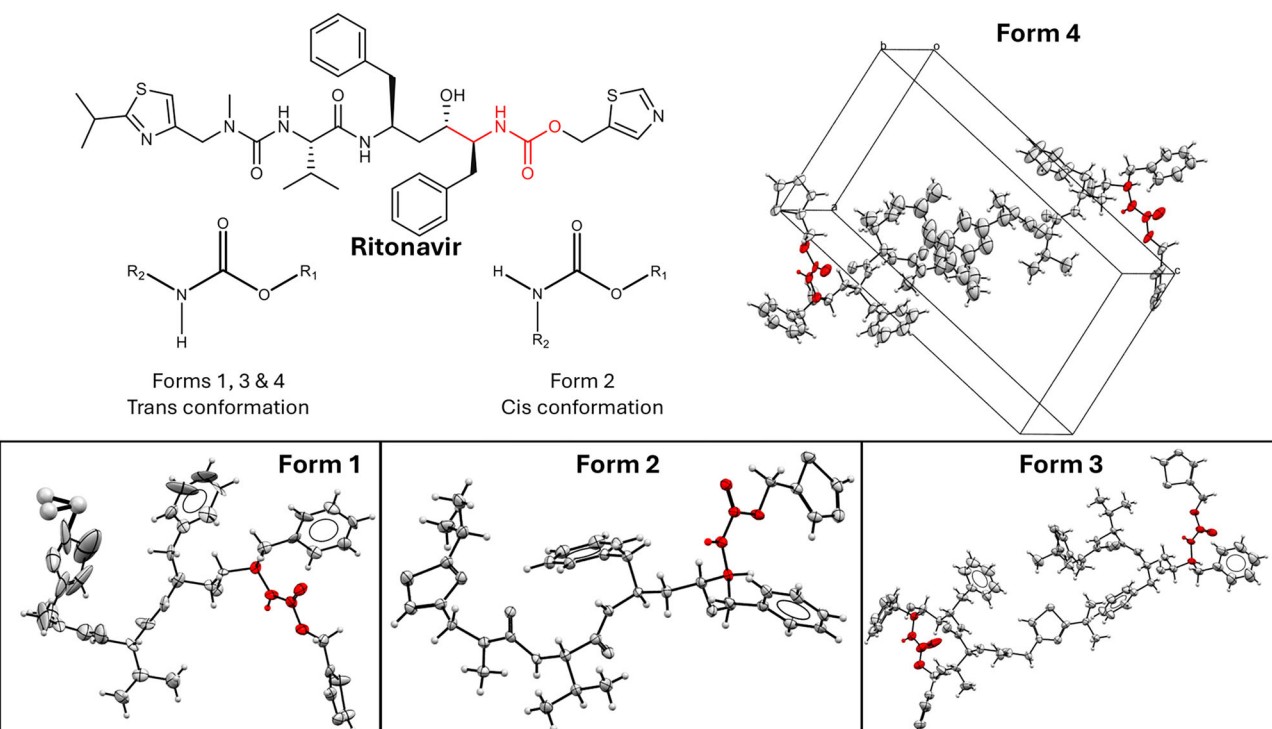

**Fig. 1 | Schematic representation of ritonavir and its polymorphs.** Two-dimensional molecular structure of ritonavir, visualization of the unit cell of the crystal structure of ritonavir form 4, and of the conformations contained in the unit cells of forms 1, 2, and 3. The carbamate configurations contained in all four forms are highlighted in red. Disorder is shown by larger thermal ellipsoids.

**Table 1 | Crystallographic information of ritonavir form 4 solved by 3D-ED. More detail can be found in Supplementary Table 2**

|  | **Ritonavir form 4** |
| --- | --- |
| **Temperature (K)** | 298 |
| **Space group** | $P2_1$ |
| **a/ Å** | 14.13 |
| **b/ Å** | 5.16 |
| **c/ Å** | 26.47 |
| **α/ °** | 90 |
| **β/ °** | 96.67 |
| **γ/ °** | 90 |
| **Volume/ Å³** | 1915.0 |

Information on the CSP predicted structure can be found in Supplementary Table 3.

finite temperature[8,31], has made it a realistic (albeit challenging) target. Furthermore, form 2 contains an energetically unfavorable *cis* conformation in the carbamate configuration of the N-methyl, N'-heterocycle motif[1]. Consequently, this form may exceed energetic cutoffs and be discarded during intermediate pruning steps due to its relative instability. Form 1 also exhibits a highly disordered fragment, indicating a significant degree of uncertainty regarding the exact atomic positions within the unit cell – an issue that is difficult to address by CSP[32]. Finally, although form 4 of ritonavir has been reported from high-throughput experiments[9], no structural data has, to the extent known to us, been published, precluding its placement on the crystal energy landscape. In this work, we aim to address both sources of incompleteness by conducting a retrospective risk assessment based on a CSP study on ritonavir, alongside leveraging a suite of cutting-edge computational and experimental techniques to accurately determine the crystal structure of form 4.

## Results and discussion
### Structure solution of form 4
Initial attempts to reproduce form 4 under precisely reported solvent and temperature conditions[9] yielded only forms 1 or 2. It was speculated that high levels of supersaturation are required to obtain form 4. We identified a specific condition, involving the use of capillary crystallization, which consistently produced mixtures of forms 1 and 4. The presence of form 4 was confirmed by both differential scanning calorimetry (DSC) and X-ray powder diffraction (XRPD). However, due to the extremely narrow experimental window required to produce form 4, obtaining a single crystal for structure determination and energy landscape placement was regarded as unlikely. Consequently, a combination of CSP, XRPD, and three-dimensional electron diffraction (3D-ED) was leveraged to solve the crystal structure of form 4. The main crystallographic information on the form 4 solution can be found in Table 1.

Form 4 exhibits a very fine needle-like morphology. When subjected to a high-energy electron beam, these particles tend to decompose rapidly, resulting in lower-resolution data. By utilizing a digitized XRPD pattern from a ritonavir solid form patent[33], we were able to find a good match with a CSP-generated crystal structure using a cross-correlation function method[34]. The corresponding coordinates were subsequently integrated with the electron diffraction data collection to obtain a complete and accurate crystal structure solution.

Interestingly, examination of this structure indicates several similarities to other unstable forms. As Fig. 1 shows, form 4 contains a *trans* carbamate configuration, like unstable forms 1 and 3, and unlike stable form 2. The hydrogen bonding network of form 4 is shown in Fig. 2. Consistent with the conformational similarity with forms 1 and 3, it has the amides forming a continuous hydrogen-bonded chain. However, there are also some notable differences. In form 1, the alcohol forms an additional intermolecular hydrogen bond with the nitrogen atom of the thiazole. In contrast, in form 4, the hydrogen bond is formed with one of the amide oxygen atoms. Meanwhile, form 3 displays a mix of the two: two of the four molecules in the

**Fig. 2 | Hydrogen bonding network in ritonavir form 4.** The hydrogen bonds are shown in purple and connect the donor-acceptor pairs identified by Mercury[71].

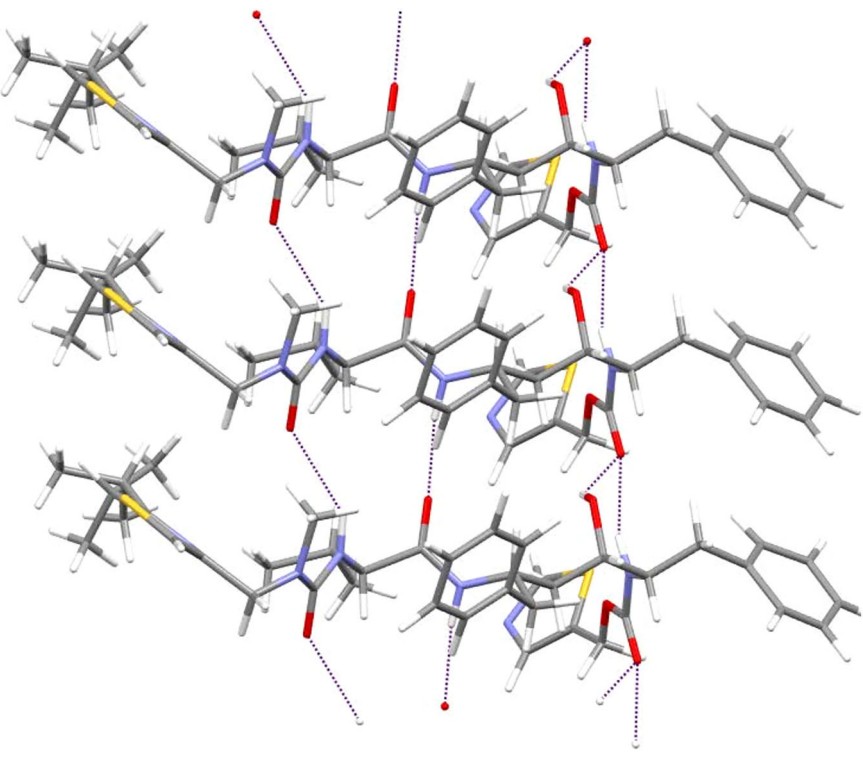

asymmetric unit create the alcohol-amide hydrogen bond seen in form 4, while the other two form the alcohol-thiazole bond found in form 1. From this analysis, it appears that the more complete hydrogen bonding pattern contained in form 2, which is responsible for its stability, can only be obtained compromising the conformational energy and introducing the unstable *cis* carbamate configuration[1].

A key feature of form 4, which is revealed by the 3D-ED enabled structural solution, is the disorder in the edge isopropyl group, which takes two configurations, with occupancies of 58% and 42% respectively. This is consistent with what we see for form 1, where these groups are also disordered, but unlike forms 2 and 3, where they are not (although thiazoles are disordered in form 3). Disorder is known to have a stabilizing effect on polymorphs[32,35], which can in some cases be modeled by computational methods[32]. This is discussed in the next sections.

**The crystal energy landscape**

The outcome of the CSP study on ritonavir is outlined in the crystal energy landscape in Fig. 2, where the relative free energies of the predicted polymorphs at room temperature (300 K) are plotted as a function of their densities. The structures matching the known forms of ritonavir are also indicated. An isolated-site disorder model[32] was used to estimate the effect of disorder on the relative free energies of forms 1 and 4.

A structure matching form 2 is the most stable predicted structure by CSP at 300 K, being 1.62 kJ·mol⁻¹ lower in energy than the disordered model of form 1 (Fig. 2). This is consistent with the experimental gap of *ca.* 2–3 kJ·mol⁻¹ at room temperature that has been reported experimentally[3], given the expected error bars of the model, which represent one standard error[8]. The lowest energy as-yet-unfound predicted polymorph of ritonavir is ranked third, being 5.51 kJ·mol⁻¹ less stable than the structure matching form 2 and 3.89 kJ·mol⁻¹ less stable than form 1 at 300 K.

From a risk assessment perspective, the initial observation of the CSP landscape is that no predicted polymorph is within one standard error of the free energy model of the structures matching forms 1 and 2. Statistically, the closest predicted crystal structure (rank 3) has a chance of about 2.0% and 0.2% of being more stable than forms 1 and 2, respectively. It is, therefore, unlikely that any more stable polymorph of ritonavir could be found, at least

within the crystallographic space covered by this CSP study, i.e., the most common chiral space groups with one molecule in the asymmetric unit cell. Thus, if this information had been known when ritonavir was being developed, and only form 1 was known, it would have shown only one, but significant, potential risk, i.e., the global minimum at 300 K, which did eventually appear as form 2 with disastrous effects. The risk due to CSP predicting a lower energy polymorph than the lead crystalline form 1, as well as the potential consequences of this predicted crystal structure emerging, would have raised significant concerns. Ritonavir is not bioavailable in the solid state, and thus the initial formulation of the drug product was a semi-solid capsule containing the drug dissolved in ethanol/water based solutions[1], which was nearly saturated with form 1[6]. Therefore, even a minor solubility decrease would have been sufficient to cause precipitation of the dissolved material and a loss in bioavailability. Examining the landscape in Fig. 3, the central estimate would be of *ca.* two-fold lower solubility for ritonavir relative to form 1 at 300 K if the global minimum had to be realized, using textbook formulae[8,25]. The solubility loss eventually realized is of similar magnitude at room temperature[6]. A structural comparison between form 1 and the CSP-predicted minimum, which eventually became form 2, would have also revealed significant differences: an analysis of the Cambridge Structural Database would have shown that the global minimum contains a rare conformation[1], but a more complete hydrogen bonding network[36], which can be a sign of a kinetically hindered polymorph[12]. Therefore, the availability of CSP data, combined with a prudent assessment of the potential risks, could have enabled an earlier mitigation of the threat. This is especially true in light of regulations that instruct pharmaceutical companies to establish clear tests and acceptance criteria if a polymorph change can affect safety or efficacy[37]. Finally, this study highlights the importance of determining disorder in crystal structures, as it stabilizes form 1, relative to ordered form 2, by 0.19 kJ·mol⁻¹. While this is insufficient to overtake the global minimum in terms of stability, it can be decisive in some cases[32].

On the other hand, optimized form 3 (YIGPIO06) is significantly unstable at 300 K, being ranked 22ⁿᵈ in Fig. 3, or 15.42 kJ·mol⁻¹ above the global minimum. The energy difference between the optimized form 3 and the structure matching form 2 is larger than what has been observed

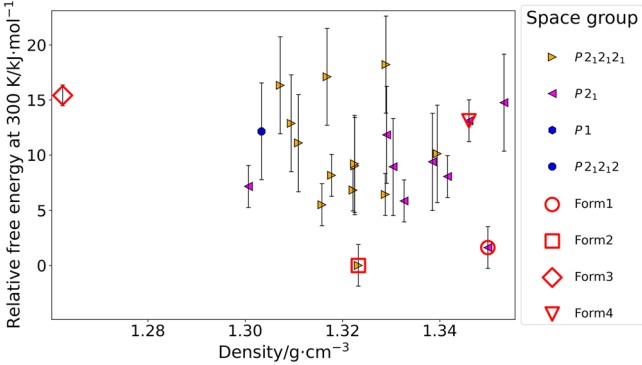

**Fig. 3 | Free energy at 300 K (relative to the global minimum) vs density plot summarizing the crystal energy landscape of ritonavir.** The free energies were calculated with the TRHu(ST)23 method. Each point on the plot corresponds to a separate computer-generated crystal structure labeled according to its space group. The structures matching form 2, the disordered models of form 1 and 4, and optimized form 3, are indicated. More details about these crystal structures can be found in Supplementary Table 3. For some predicted polymorphs, the error bar, representing one standard error, is significantly larger, as only the lattice energy was explicitly calculated for those, while the vibrational component to free energy was only estimated.

experimentally from solubility ratios (*ca.* 8 kJ·mol$^{-1}$)[3]. However, all four molecules in the asymmetric unit of form 3 are disordered, suggesting that the selected representative configuration may not be the most thermodynamically stable one. Furthermore, disorder could significantly stabilize the overall energy of form 3, especially because the disordered groups are closely interacting[32]. Unlike for forms 1 and 4, we did not make an attempt to estimate the energetic impact of disorder on form 3. This is because the Z' = 4 (i.e., four symmetry independent molecule in the unit cell) nature of this polymorph, with all four molecules being disordered, and with the disordered groups closely interacting, drastically increases the number of possible combinations that would need to be studied[32], making an estimate computationally expensive. Furthermore, the additional benefit of this calculation would be minimal as form 3 is already known experimentally to be significantly unstable[3].

Finally, form 4, determined in this work for the first time (as far as we are aware), is ranked 20$^{th}$, 13.11 kJ·mol$^{-1}$ less stable than the global minimum. Its solution closes a significant gap in our understanding of the polymorphism of ritonavir and conclusively proves its significant thermodynamic instability relative to forms 1 and 2. Furthermore, this solution shows the importance of the disordered nature of form 4, which is estimated to stabilize it by 0.58 kJ·mol$^{-1}$ at 300 K; although the starting energy gap is too large for disorder to make form 4 competitive with its more stable counterparts, understanding and calculating disorder could be fundamental in more ambiguous cases[32].

**What are the implications of this study for the pharmaceutical industry?**

This CSP study shows that the most notorious polymorphic disaster in the recent drug development history was predictable. The computational cost associated with running it (see Supplementary Note 4.2 for detail) would have been inconsequential to the eventual price of failure, estimated at 250 million USD[5]. CSP is complementary to experimental work as it can help understand if a known structure is thermodynamically stable or the product of favorable crystallization kinetics[20]. However, computational modeling has weaknesses: our understanding and ability to model nucleation and growth kinetics is poor[38,39], and therefore, there is a significant gap between CSP predicting alternative polymorphs and our ability to realize and characterize them[26,38,40–43]. For ritonavir, CSP would have forecast the existence of a more stable polymorph that was likely to be kinetically hindered because of its structural characteristics, and shown that it would have been a

significant threat due to a potential solubility loss larger than what the formulated product could tolerate. Some experiments could have been suggested, e.g., choosing solvents more likely to favor the conformation of the putative more stable polymorph of ritonavir[44] or crystallizing it under low supersaturation conditions[45]. At a minimum, the risks outlined by the CSP landscape would have led to performing high-throughput crystallization experiments, leading to an earlier discovery and characterization of form 2. A retrospective high-throughput crystallization screen on ritonavir consisting of ~2000 experiments indeed confirms that form 2 can be produced by the sheer number of attempts, even if not specifically targeted towards it[9]. A less brittle formulation, able to accommodate the form 2 solubility, would then have been selected[6] and the disaster would have been averted.

Another weakness of CSP that this study reveals is that it cannot realistically explore the whole crystallographic space. While Z'= 1 CSP can be performed accurately and routinely, Z' = 2 CSP would be difficult and likely not realistically completed to a full convergence for a molecule of the size of ritonavir, with Z' > 2 being currently unfeasible. Therefore, CSP would have not been able to predict the existence of form 3 of ritonavir with Z' = 4. Although form 3 happens to be very unstable, crystal structures with large Z' can be the thermodynamically stable polymorphs[46]. Therefore, a proper risk management dictates that some screening work is needed even when no threat is predicted by CSP. This limitation is even more evident for solvates, given the plethora of possible stoichiometries and the possibility of forming non-stoichiometric adducts[46–48].

This study also proves the power of 3D-ED in enabling the solution of crystal structures when single crystals of sufficient dimensions cannot be grown[49,50]. Ritonavir form 4 could not be solved by the "gold standard" of single-crystal XRD, but a combination of 3D-ED with computational methods and experimental XRPD data produced an accurate solution and showed it is significantly less stable than forms 1 and 2[49]. Being able to solve crystal structures for which single-crystal XRD is not an option is important from a risk assessment perspective. Often, high throughput experimentation finds new crystalline forms that are just seen as peaks on XRPD patterns, without the possibility of isolating them, solving their structures, and verifying their relative stabilities. This can cause significant concerns in drug development. Ritonavir form 4 ended up being unstable, but there are cases in which solving a crystal structure from a small sample size and therefore determining its relative thermodynamic stability is fundamental to quantify and assess risks. In this scenario, leveraging less traditional methods like 3D-ED or a combination of CSP and experimental data[51–53] is the only option. This ability to characterize crystal structures from a small sample size can also have intellectual property implications[54].

Finally, the work reveals the importance of determining structural disorder in crystal structures and quantifying its effects on the stability of polymorphs for a complete risk assessment on the polymorphic landscape of drug candidates. Disorder has a modest but not irrelevant stabilizing effect on ritonavir form 1 and particularly on form 4, and likely on form 3 too (where the effect was not computed), relative to ordered form 2. Crystalline disorder and its stabilizing effect are getting increasingly recognized in the solid-state community, and this study shows that this interest is well deserved.

In conclusion, we were able to show that the ritonavir disaster would be predictable with state-of-the-art computational workflows: had they been available, a risk assessment based on in silico results would have likely avoided its occurrence. Furthermore, this study was able to solve and reveal the disordered nature of ritonavir form 4, via a combination of 3D-ED, experimental XRPD data, and computational modeling. Form 4 is shown to be significantly less stable than forms 1 and 2 of ritonavir and therefore not a threat to their development. This method for structural solution and disorder determination could be fundamental to perform risk assessment in several cases where polymorphs cannot be determined by single crystal XRD methods.

## Methods

### Experimental forms of ritonavir

Four single-component polymorphs of ritonavir have been reported in the literature. Forms 1 and 2 are the main protagonists of the original ritonavir story, and both have well-determined crystal structures deposited in the Cambridge Structural Database (CSD)[55]. This paper considers crystal structures with reference codes YIGPIO02[1] and YIGPIO03[1] as the best representatives of forms 1 and 2, respectively. YIGPIO02 has a missing hydroxyl proton and positional disorder around an isopropyl motif, with the two orientations (major and minor) having 56% and 44% occupancies, respectively[1]. The experimental stability of form 2 relative to form 1 has been reported to correspond to *ca.* 2–3 kJ·mol$^{-1}$ at room temperature (300 K), using solubility ratios[3].

More recently, a new polymorph of ritonavir (form 3) has been reported[56,57]. Form 3 has been described simultaneously in two independent publications by Li et al.[2] and Yao et al.[3], and it appears to be heavily disordered. Yao et al. deposited two solutions of form 3 in the CSD: YIGPIO04[3] has two molecules of ritonavir in the asymmetric unit cell (i.e., Z' = 2) and is in the *C2* space group, while YIGPIO05[3] is Z' = 1 and also in the *C2* space group; data was collected at different temperatures. Li et al. have deposited YIGPIO06[2], a Z' = 4 crystal structure in the *P1* space group, collected using synchrotron data. After careful analysis, we decided to consider YIGPIO06 as the best representative of form 3 as it is a more realistic model for this polymorph; see Supplementary Note 1 for a detailed justification. Form 3 appears to be significantly unstable thermodynamically, with a free energy gap of *ca.* 8 kJ·mol$^{-1}$ relative to form 2 and *ca.* 6 kJ·mol$^{-1}$ relative to form 1 from solubility ratios[3].

A fourth polymorph has been identified, which will be referred to as form 4[9,33]. This true polymorph of ritonavir was reported as non-solvated and metastable[9]. Initially, it was prepared only from acetonitrile-acetate solvent mixtures, but more recently, it has also been obtained in salt cocrystal attempts with oxalic acid[58]. To date, no full 3D solution is available, but only an X-ray powder diffraction (XRPD) pattern and thermal analysis[33].

### Crystal structure prediction study

The CSP study on ritonavir was performed using GRACE 2.8[59]. The process is described in more detail in Supplementary Note 2. First, a tailor-made force field (TMFF)[60] specific to ritonavir was fitted from scratch; then, crystal structures were generated in 21 chiral space groups with Z' = 1. Although in most industrial applications Z' = 2 crystal structures are also considered, as they are especially prevalent for chiral compounds[46], we deliberately decided to limit the CSP to Z' = 1 given the substantial cost associated with working on such a large and flexible molecule. Note that ritonavir was treated as fully flexible during the crystal structure generation, avoiding the need to perform a costly conformational search and of arbitrarily reducing the conformational search space. Subsequently, the most promising predicted polymorphs underwent quantum-mechanical optimizations at the PBE-NP[61,62] level of theory. Finally, all the low-energy crystal structures from the CSP study had their lattice energy calculated with the highly accurate and validated TRHu(ST)23 (temperature- and relative-humidity-dependent free-energy calculations with standard deviations)[8] method available through GRACE 3.1. This method involves recalculating the crystalline energies at the PBE0-MBD-NL[63–65] level of theory and adding a post-Hartree Fock monomer correction computed with the MP2D method[66]. For a subset of 13, including the most stable predicted polymorphs in terms on lattice energy and those matching experimentally known forms of ritonavir, the vibrational component to free energies ($F_{vib}$)[31,67] was also calculated within the harmonic approximation. For the remaining less stable predicted polymorphs, given the high computational associated with the phonon frequency calculations, finite temperature $F_{vib}$ was estimated by applying a uniform shift to all lattice energy values based on the average $F_{vib}$ for the 13 for which it had been explicitly calculated (see the Supplementary Note 4.1 for details), which is common practice in similar situations[24,68]. The method outlined by Firaha et al.[8] was finally used to compute the error bar in terms of relative free energy associated with each predicted polymorph of ritonavir. More detail can be found in the Supplementary Note 4.3, but in summary the error bars represent one standard error derived from the discrepancies between experimental and computed free energy differences estimated from a thorough benchmarking against known data; furthermore, for the crystal structures for which $F_{vib}$ was not explicitly calculated, but estimated by applying uniform shifts, the standard deviation between their lattice and free energies at room temperature needs to be included in the error bar. In these cases, the overall confidence interval will be larger.

Note that since our preferred solution of form 3 is a Z' = 4 structure, it could not be found by the Z' = 1 crystal structure generation and was therefore optimized independently with the TRHu(ST)23 method to calculate its free energy at finite temperature and place it on the crystal energy landscape. Given the high level of disorder and the interacting disordered groups, which would require more complex models to estimate its thermodynamic effects[32,69], we decided only to optimize the major disordered component. This may cause a less accurate energy estimation due to potentially not choosing the most stable disordered configuration at that level of theory and neglecting the stabilizing effect of configurational entropy.

### Structure solution of ritonavir form 4

To solve the crystal structure of form 4 of ritonavir, we leveraged a combination of the XRPD pattern, the CSP-generated polymorphs, and three-dimensional electron diffraction (3D-ED) analysis. First, we compared the 'digitized' experimental XRPD pattern against the CSP-generated crystal structures using a cross-correlation factor method[34] available in GRACE. The outcome of this analysis is visualized in Supplementary Fig. 4, with the crystal structure ranked 20[th] in terms of lattice energy having the best score. In parallel, we prepared a mixture form 1 and form 4 material by capillary crystallization[70] (see Supplementary Note 3.1) and performed 3D-ED on a sample of 13 crystals of form 4 to obtain a solution of this crystal structure. Although 3D-ED successfully yielded unit cell and space group information, the ab initio structure solution initially failed, but was eventually successful when using the best CSP-generated matching structure as starting point. More detail on the preparation of form 4 and the 3D-ED analysis is discussed in the Supplementary Note 3.

## Data availability

The structure files of the computationally generated crystal structures, in the format of Crystallographic Information Files (.cif), are available in Supplementary Data 1, named by their ranks in Supplemntary Table S3. Crystallographic data for ritonavir form 4 has been deposited at the Cambridge Crystallographic Data Center, under deposition number CCDC 2411739; it is also included here in Supplementary Data 2. This data is provided free of charge by the joint Cambridge Crystallographic Data Center and Fachinformationszentrum Karlsruhe Access Structures service and can be accessed at www.ccdc.cam.ac.uk/structures.

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

## Acknowledgements

The authors thank Marcus Neumann and Avant-Garde Materials Simulation for developing and supporting the GRACE crystal structure prediction software and the CCDC for curating the CSD. Mohammad T. Chaudhry and Andrew W. Kelly acknowledge support from Merck & Co., Inc. (Rahway, NJ, USA) in the form of a post-doctoral fellowship.

## Author contributions

L.I., A.W.K. and A.Y.L. designed research; L.I., A.W.K. and C.J. performed research; C.J. and D.S. contributed analytical tools; L.I., A.W.K., M.T.C., A.Y.L. and C.J. analyzed data and wrote the paper.

## Competing interests

L.I., A.W.K., M.T.C., and A.Y.L. are employees of Merck Sharp & Dohme LLC, a subsidiary of Merck & Co., Inc., Rahway, NJ, USA, and potentially own stock and/or hold stock options in Merck & Co., Inc., Rahway, NJ, USA. All other authors declare no competing interests.
