## [Transparent Peer Review file · Communications Chemistry]

Predicting the ritonavir crisis by revisiting the polymorph landscape with crystal structure prediction and form 4 structure solution

Corresponding Author: Dr Luca Iuzzolino

Version 0:

Reviewer comments:

Reviewer #1

(Remarks to the Author)

This study successfully demonstrates crystal structure prediction (CSP) for Ritonavir and determines Form IV's structure. However, this application does not demonstrate significant advances in techniques, as the approaches have been widely used in the field with numerous existing cases. Additionally, the risk assessment conclusions remain unclear regarding success or failure, since the discussion only addresses the possibility of obtaining Form II. Further study is required to establish clear conclusions.

Recommendation: Major Revision

The manuscript requires a complete risk assessment narrative with clear conclusions before it can be considered suitable for publication.

Reviewer #2

(Remarks to the Author)

The paper involving CSP to investigate the polymorphism of ritonavir is somewhat disappointing and places too much weight on a belated CSP prediction and stability ranking, whereas the problems dealing with disorder and the resolution of form 4 also involving disorder are much more current and deserve much more attention. Moreover, the paper lacks a clear outline, meandering between several subjects.

The real feat of the paper, which is in our opinion buried in the SI, is the solution of form 4, which is complicated due to disorder. Disorder being an element that is taken more and more seriously as a stabilising factor, difficult to model in CSP and therefore hard to rank on an energy scale. The structure resolution of form 4 in combination with the role played by disorder should get more emphasis in the paper, while the inevitable conclusion that we know better now about forms I and II is simply not new.

In this respect, the discussion about what the CSP result for form II would trigger is somewhat superfluous. Any form lower in energy than the experimentally observed form will trigger laboratory experiments to try to find it, whether the unfound form seems kinetically hindered or not, it is simply a risk once it is known it is out there, also considering the existing regulations on polymorph screening that should be cited in the paper: "Specifications: Test Procedures And Acceptance Criteria For New Drug Substances And New Drug Products: Chemical Substances - Q6A, International conference on harmonisation of technical requirements for registration of pharmaceuticals for human use, 6 October 1999."

Thus, at present the paper is written in a way that it is not acceptable for commschem, however, if the paper is revised in which the text is streamlined (also the SI) and the solution of form 4 in combination with its CSP analysis is made into the core of the paper, this may in fact be a paper of interest for commchem.

Other points to improve the paper

Please do not use metastable in a comparative degree. Metastable means in general that the state is kinetically locked in and less stable than a more stable state. Using expressions such as "Very metastable" is confusing as it may mean a low or high kinetic barrier or a large or small Gibbs free energy difference: simply write that a given metastable state is very unstable or has a high Gibbs free energy in relation to the stable form.

Abstract: "pharmaceutical polymorphism", too much of a contraction, I would rephrase this as "structural polymorphism in pharmaceuticals...".

Table 1 data differs from Table S2 for column 2 lines 4, 6 and 10. The last digit is an uncertainty or not? please clarify.

In section "structure solution of form 4" a figure with a comparison of the crystal structure of forms 1,2, and 3 vs the newly obtained form 4 would be appreciated as the paragraph following figure 1 discusses the differences in the crystal structures of forms 1 and 3.

Figure 1. the structure is not very clear, is there a better way to show this structure with for example only two molecules highlighting the hydrogen bonds?

page 6: first line: "a match to form 2..."

Clearer would be "The structure matching form 2..."

also in other occasions, replace "the match" by "the matching structure" (2nd paragraph p6 for example)

page 6: extremely metastable; confusing terminology.

page 6, first paragraph, what is the crystallographic space covered by the CSP study (methods and SI are placed later or elsewhere)?

page 6, second paragraph:

"The gap between optimized form 3 and the match to form 2 is significantly larger than what has been seen experimentally from solubility ratios"

sentence not clear, 'the energy difference between the optimised form 3 and the structure matching form 2'?

page 6, second paragraph: "it is a very metastable polymorph, "

do not use metastable in a comparative degree, terms such as stable - unstable suffice.

page 6, second paragraph "like form 3, would also be irrelevant from a pharmaceutical development perspective as it will not introduce significant solid-state stability risks."

although this may be true, this cannot be concluded from a CSP study alone. transformation can be kinetically hindered and the form could possibly be developed as a more soluble polymorph.

Section "The crystal energy landscape": Page 7 line 22: About CSP in support of the development of Ritonavir, it is not clearly stated that CSP was little developed and little used at the end of the 20st century and therefore irrelevant to the stability study of Ritonavir.

Section "What does this study show about CSP and its application in the pharmaceutical industry?"

The discussion that follows is very broad and meanders into subjects that have little relation with the CSP of ritonavir; this should be shortened and made relevant in relation to the subject matter.

Page 9, line 3 : the purpose "is just the case for single-component crystal structures: for solvates the situation can become even more complicated given the plethora of possible stoichiometries and the possibility of forming non-stoichiometric adducts." is written twice with slightly different references.

No TOC graphic is present.

SI

Concerning the resolution of form 4 from a powder sample containing forms 1 and 4, the description is not very clear. It may in fact be better not to divide the sample preparation and the 3DED experiments. Moreover, it should be more clearly indicated how the sample was treated and how the sample was analysed with the 3DED equipment. Mention is made of 135 crystals being measured, whereas only 13 appear in the table S1. What is the observed distribution of forms 1 and 4? How were the 13 crystals selected?

In table S2, rhocalc (in which rho is greek letter), calc should be subscript for clarity, a space is missing between g et cm⁻³.

Page 13, line 3 : "to correspond" is written twice.

Reviewer #3

(Remarks to the Author)

I co-reviewed this manuscript with one of the reviewers who provided the listed reports. This is part of a Communications Chemistry initiative to facilitate training in peer review and to provide appropriate recognition for Early Career Researchers who co-review manuscripts.

Reviewer #4

(Remarks to the Author)

This is a nice paper, which demonstrates how crystal structure prediction methods might have been able to anticipate the appearance of a second polymorph of ritonavir. This is an important example because ritonavir is often quoted as a motivation for the development of CSP methods for pharmaceuticals. Therefore, I think that the paper is important for the field: it answers the important question about whether CSP could have helped avoid the whole ritonavir story of late-appearing polymorphism and its dramatic consequences. I support publication of the manuscript, but have some comments that I think should be addressed:

p.3 (introduction): "In this work, we revisit the ritonavir story by advanced computational means by a) performing a CSP under Blind Test conditions"

Is this accurate? I don't agree. "Blind test conditions" means, for most readers, to perform predictions without knowledge of the known structures. The case under study is a situation where form 1 and 2 are known; it is not possible to perform predictions for this molecule under true blind test conditions. The whole reason for the organisation of blind tests in crystal structure prediction is that success reported in the literature for crystal structure prediction methods was being assessed on molecules where the crystal structures were already known, and so where it is possible that the known structure influences, even in a subconscious way, how the prediction calculations are performed. Therefore, "blind test conditions" should be reserved for studies where calculations are performed before the structures are known. The term should not be used to mean "we did not intentionally use any experimental information in setting up the calculations."

p.6 "The lowest energy as-yet-unfound predicted polymorph of ritonavir is CSP rank 4, which is 5.51 kJ·mol⁻¹ less stable than the match to form 2 and 3.89 kJ·mol⁻¹ less stable than the disordered model of form 1 at 300 K."

The lowest energy predicted structure corresponds to form 2, the next structure to form 1. Why is the rank 4 structure the lowest energy unobserved structure? What about the rank 3 structure from the CSP study? Surely, it is lower energy than rank 4 and seems to be an unobserved structure.

Figure 2: Please state in the caption more information about the meaning of the error bars. The Methods section states "The method outlined by Firaha et al. 8 was finally used to compute the error bar in terms of relative free energy associated with each predicted polymorph of ritonavir." Are these the standard deviation calculations described in reference 8? If the error bars in the figure are 1 standard deviation, is the statement (page 6) "Note that no predicted polymorph is within the confidence interval of the model of the matches to the known most stable forms" correct? Is this statistically justified based on the observed non-overlap of one standard deviation? Some simple statistical calculations could assess the likelihood, based on these error estimates, that one of the higher energy structures is at or lower in energy than the matches to forms 1 and 2.

p.9. There is a repeated sentence at the top of this page: "This is just the case for single component crystal structures: for solvates, the situation can become even more complicated given the plethora of possible stoichiometries and the possibility of forming non-stoichiometric adducts. 45-47 And this is just the case for single- component crystal structures: for solvates the situation can become even more complicated given the plethora of possible stoichiometries and the possibility of forming non-stoichiometric adducts. 46,47"

p.9. In discussing solvates and the challenges that they create for CSP, "Although methods have been developed to help with these cases, 48 they require CSP to predict a solvent-free scaffold..." Reference 48 presents a method for hydrates. More general work on solvates was presented in Chemistry—A European Journal 2009, 15 (47), 13033-13040.

p.9. The point about combining CSP with less traditional methods, such as electron diffraction, is important. A important aspect of ED is the small sample size requirement, which means that small amounts of new polymorphs might be able to be identified within the bulk sample of a known form, as shown in Chem. Eur. J., 19: 7883-7888. <https://doi.org/10.1002/chem.201204369>, where a polymorph was identified and structure proposed by comparison to CSP. This has implications in the discussion of risk assessment, and potentially also for intellectual property.

Version 1:

Reviewer comments:

Reviewer #1

(Remarks to the Author)

After the major revision, the manuscript clearly demonstrates how CSP can effectively derisk the unexpected polymorphism issue in the well-known Ritonavir case, and has successfully solved the previously uncharacterized Form 4 structure through CSP and ED techniques. The paper is recommended for acceptance ending minor revisions, with detailed comments provided in the attached document.

Reviewer #2

(Remarks to the Author)

The paper has been revised into a much better version of itself and can be published in this form.

Reviewer #3

(Remarks to the Author)

I co-reviewed this manuscript with one of the reviewers who provided the listed reports. This is part of the Communications Chemistry initiative to facilitate training in peer review and to provide appropriate recognition for Early Career Researchers who co-review manuscripts.

Reviewer #4

(Remarks to the Author)

I have read the response to reviews and the revised manuscript. My opinion is that the authors have satisfactorily addressed the criticisms from the review of the original manuscript. The revised manuscript is an important contribution to the field, showing that current crystal structure prediction methods are able to address systems like Ritonavir and make predictions that could have averted the problems of late appearing polymorphism. I support publication.

Version 2:

Reviewer comments:

Reviewer #1

(Remarks to the Author)

I have reviewed the authors' response and the revised manuscript. The authors have adequately addressed most of the points. One minor suggestion would be to improve readability and methodological transparency of the revised sentence on Page 13, for example by explicitly stating that "...the finite-temperature F_{vib} was estimated by applying a uniform shift to all values based on the average F_{vib} from the 12 calculated crystals...(see SI Section 4.1 for details)". Should this suggestion be incorporated, I would be pleased to recommend the manuscript for publication.

Reviewers' comments:

Reviewer #1 (Remarks to the Author):

This study successfully demonstrates crystal structure prediction (CSP) for Ritonavir and determines Form IV's structure. However, this application does not demonstrate significant advances in techniques, as the approaches have been widely used in the field with numerous existing cases. Additionally, the risk assessment conclusions remain unclear regarding success or failure, since the discussion only addresses the possibility of obtaining Form II. Further study is required to establish clear conclusions.

Recommendation: Major Revision

The manuscript requires a complete risk assessment narrative with clear conclusions before it can be considered suitable for publication.

We thank Reviewer #1 for the constructive criticism, which we believe has been addressed. We have substantially revised the manuscript, which now has a much clearer risk assessment narrative focused on how performing crystal structure prediction (CSP) would have changed the ritonavir story, and on the importance of solving and calculating the relative stability of form 4:

- CSP would have flagged a significant risk to the drug product well before form 2 appeared, due to the structural characteristics pointing to a kinetically hindered polymorph, and to the low tolerance of the formulation for any solubility loss.
- The methods described in this work are the only ones that have been able to solve and place on the crystal energy landscape form 4, proving its instability.

We also believe we have a clear conclusion:

- CSP would have prevented the ritonavir disaster.
- Form 4 is unstable and therefore not a significant issue from a development standpoint.

Reviewer #2 (Remarks to the Author):

The paper involving CSP to investigate the polymorphism of ritonavir is somewhat disappointing and places too much weight on a belated CSP prediction and stability ranking, whereas the problems dealing with disorder and the resolution of form 4 also involving disorder are much more current and deserve much more attention. Moreover, the paper lacks a clear outline, meandering between several subjects.

The real feat of the paper, which is in our opinion buried in the SI, is the solution of form 4, which is complicated due to disorder. Disorder being an element that is taken more and more seriously as a stabilising factor, difficult to model in CSP and therefore hard to rank on an energy scale. The structure resolution of form 4 in combination with the role played by disorder should get more emphasis in the paper, while the inevitable conclusion that we know better now about forms I and II is simply not new.

In this respect, the discussion about what the CSP result for form II would trigger is somewhat superfluous. Any form lower in energy than the experimentally observed form will trigger laboratory experiments to try to find it, whether the unfound form seems kinetically hindered or not, it is simply a risk once it is known it is out there, also considering the existing regulations on polymorph screening that should be cited in the paper: "Specifications: Test Procedures And Acceptance Criteria For New Drug Substances And New Drug Products: Chemical Substances - Q6A, International conference on harmonisation of technical requirements for registration of pharmaceuticals for human use, 6 October 1999."

Thus, at present the paper is written in a way that it is not acceptable for commschem, however, if the paper is revised in which the text is streamlined (also the SI) and the solution of form 4 in combination with its CSP analysis is made into the core of the paper, this may in fact be a paper of interest for commchem.

We thank Reviewer 2 for the criticism of this paper, which we believe we have addressed. In this substantially revised manuscript we have put more emphasis on the form 4 solution, the risk assessment that it enables by proving its instability, and the importance of using 3D-ED combined with computational methods where there is a lack of suitable single crystals for structural determination. We have also put more focus on the importance of solving the disorder of crystal structures, due to the stabilizing effect of disorder relative to ordered polymorphs, and on how computational methods can estimate this effect. We have also shortened and streamlined both the manuscript and the SI, trying to keep the discussion more focused and to the point and giving both a clearer outline.

On the point regarding the work that the CSP results would trigger, we have emphasized why these CSP results would have been viewed as especially concerning: due the low tolerance of the formulation for a loss in solubility, the existence of an unfound, more stable global minimum (which eventually appeared as form 2) would have been seen as particularly problematic from a risk assessment perspective, especially given regulations that mandate strict tests and acceptance criteria if the polymorph change can cause loss of safety or efficacy (we have cited the specifications mentioned above). However, while we have de-emphasized it, we do not believe that the structural factors pointing to a kinetic hindrance of the global free energy minimum/form 2 should be totally discounted: given the relative low energy difference between the rank 1/form 2 structure and form 1, well within the expected standard error of the energy model, the amount of worry this would cause (and the associated screening work) would be less if the global minimum were to be structurally and conformationally similar to form 1, as this would be seen as making it unlikely that it has not yet been experimentally realized due to a kinetic barrier.

Other points to improve the paper

Please do not use metastable in a comparative degree. Metastable means in general that the state is kinetically locked in and less stable than a more stable state. Using expressions such as "Very metastable" is confusing as it may mean a low or high kinetic barrier or a large or small Gibbs free energy difference: simply write that a given metastable state is very unstable or has a high Gibbs free energy in relation to the stable form.

We have removed the word 'metastable' and replaced it throughout the manuscript with the words 'unstable' or 'instability'. The only exception is when we discuss form 4 from the previous literature where it was explicitly defined as metastable (Morissette et al. (2003) Proc. Natl. Acad. Sci. USA, 10.1073/pnas.0437744100).

Abstract: "pharmaceutical polymorphism", too much of a contraction, I would rephrase this as "structural polymorphism in pharmaceuticals...".

This has been rephrased as suggested by the reviewer.

Table 1 data differs from Table S2 for column 2 lines 4, 6 and 10. The last digit is an uncertainty or not? please clarify.

The digits between brackets in Table S2 are the standard deviation, which is reported in Table S2 but not in Table 1 for brevity. E.g. 5.16(13) signifies a unit cell of 5.16 Å with a standard deviation of 0.13 Å. This notation is also used in Mercury.

In section "structure solution of form 4" a figure with a comparison of the crystal structure of forms 1,2, and 3 vs the newly obtained form 4 would be appreciated as the paragraph following figure 1 discusses the differences in the crystal structures of forms 1 and 3.

We have added a new Figure 1 where the unit cell of form 4 is shown, with an emphasis on disorder and on the *trans* carbamate configuration. Figure 1 also shows the ritonavir conformations contained in forms 1, 2, and 3, once again emphasizing disorder and the carbamate configurations. Including the whole unit cells of forms 1, 2, and 3 would have made the image too cluttered in our view; this is why for the previously known forms we only included the conformations.

Figure 1. the structure is not very clear, is there a better way to show this structure with for example only two molecules highlighting the hydrogen bonds?

We have moved the original image to Figure 2, and only included three molecules with the associated H-bonds, to show the chained-nature of the motif.

page 6: first line: "a match to form 2..."

Clearer would be "The structure matching form 2..."

also in other occasions, replace "the match" by "the matching structure" (2nd paragraph p6 for example)

This has been rephrased throughout both the manuscript and the SI.

page 6: extremely metastable; confusing terminology.

Fixed, as mentioned above.

page 6, first paragraph, what is the crystallographic space covered by the CSP study (methods and SI are placed later or elsewhere)?

While it is mentioned in the SI and the Methods, we have added a clarification: "at least within the crystallographic space covered by this CSP study, *i.e.* the most common chiral space groups with one molecule in the asymmetric unit cell"

page 6, second paragraph:

"The gap between optimized form 3 and the match to form 2 is significantly larger than what has been seen experimentally from solubility ratios"

sentence not clear, 'the energy difference between the optimised form 3 and the structure matching form 2'?

This has been rephrased as suggested by the reviewer.

page 6, second paragraph: "it is a very metastable polymorph, "
do not use metastable in a comparative degree, terms such as stable - unstable suffice.

Fixed, as mentioned above.

page 6, second paragraph "like form 3, would also be irrelevant from a pharmaceutical development perspective as it will not introduce significant solid-state stability risks."

although this may be true, this cannot be concluded from a CSP study alone. transformation can be kinetically hindered and the form could possibly be developed as a more soluble polymorph.

We agree with this comment and have therefore removed that sentence.

Section "The crystal energy landscape": Page 7 line 22: About CSP in support of the development of

Ritonavir, it is not clearly stated that CSP was little developed and little used at the end of the 20st century and therefore irrelevant to the stability study of Ritonavir.

In the Introduction, we added a sentence: “Although CSP was not fully developed in the 1990s”. However, we do not believe the landscape is irrelevant, since our aim is to prove that CSP would have prevented the ritonavir disaster by flagging a significant threat (which was eventually realized) to form 1, not just to prove that form 1 is less stable than form 2, which we agree it is a well-established fact.

Section “ What does this study show about CSP and its application in the pharmaceutical industry?”

The discussion that follows is very broad and meanders into subjects that have little relation with the CSP of ritonavir; this should be shortened and made relevant in relation to the subject matter.

We have renamed this section “What are the implications of this study for the pharmaceutical industry?”, shortened it, and made it more focused on the results of this paper and its implications for drug development.

Page 9, line 3 : the purpose “is just the case for single-component crystal structures: for solvates the situation can become even more complicated given the plethora of possible stoichiometries and the possibility of forming non-stoichiometric adducts.” is written twice with slightly different references.

We have removed this repetition.

No TOC graphic is present.

To our knowledge, Communications Chemistry submissions do not include a TOC graphic (https://mts-commschem.nature.com/cgi-bin/main.plex?form_type=display_auth_instructions).

SI

Concerning the resolution of form 4 from a powder sample containing forms 1 and 4, the description is not very clear. It may in fact be better not to divide the sample preparation and the 3DED experiments.

It is common practice in the experimental section to have preparation and analytics separated. The sample prepared in this way was not only used for 3D-ED, but also for other analytical techniques like powder XRD and DSC, therefore this division makes sense and is clearer to read in our view. However, we have put both these as subsections of the same SI section 3, which discusses the structure solution of form 4.

Moreover, it should be more clearly indicated how the sample was treated and how the sample was analysed with the 3DED equipment.

There was no further sample treatment (like suspending or grinding), the sample was used as received and dispersed on the sample support, this has been made clearer in the Supporting Information.

Mention is made of 135 crystals being measured, whereas only 13 appear in the table S1. What is the observed distribution of forms 1 and 4?

Of 135 crystals 17 had the unit cell of form 4, this number was added in the Supporting Information (the majority was form 1 as already stated, other forms were not identified).

How were the 13 crystals selected?

The 13 best datasets were selected based on R_{int} values during merging, this has been added to the Supporting Information and these are the ones shown in the table.

In table S2, rhocalc (in which rho is greek letter), calc should be subscript for clarity, a space is missing between g et cm⁻³.

This has been fixed.

Page 13, line 3 : “to correspond” is written twice.

This has been fixed.

Reviewer #3 (Remarks to the Author):

I co-reviewed this manuscript with one of the reviewers who provided the listed reports. This is part of a Communications Chemistry initiative to facilitate training in peer review and to provide appropriate recognition for Early Career Researchers who co-review manuscripts.

We thank Reviewer 3.

Reviewer #4 (Remarks to the Author):

This is a nice paper, which demonstrates how crystal structure prediction methods might have been able to anticipate the appearance of a second polymorph of ritonavir. This is an important example because ritonavir is often quoted as a motivation for the development of CSP methods for pharmaceuticals. Therefore, I think that the paper is important for the field: it answers the important question about whether CSP could have helped avoid the whole ritonavir story of late-appearing polymorphism and its dramatic consequences. I support publication of the manuscript, but have some comments that I think should be addressed:

We thank Reviewer 4 for supporting the publication of this manuscript.

p.3 (introduction): “In this work, we revisit the ritonavir story by advanced computational means by a) performing a CSP under Blind Test conditions”

Is this accurate? I don't agree. “Blind test conditions” means, for most readers, to perform predictions without knowledge of the known structures. The case under study is a situation where form 1 and 2 are known; it is not possible to perform predictions for this molecule under true blind test conditions. The whole reason for the organisation of blind tests in crystal structure prediction is that success reported in the literature for crystal structure prediction methods was being assessed on molecules where the crystal structures were already known, and so where it is possible that the known structure influences, even in a subconscious way, how the prediction calculations are performed. Therefore, “blind test conditions” should be reserved for studies where calculations are performed before the structures are known. The term should not be used to mean “we did not intentionally use any experimental information in setting up the calculations.”

We agree with the comment, and have therefore removed the reference to “Blind Test conditions”.

p.6 “The lowest energy as-yet-unfound predicted polymorph of ritonavir is CSP rank 4, which is 5.51 kJ · mol⁻¹ less stable than the match to form 2 and 3.89 kJ · mol⁻¹ less stable than the disordered model of form 1 at 300 K.”

The lowest energy predicted structure corresponds to form 2, the next structure to form 1. Why is the rank 4 structure the lowest energy unobserved structure? What about the rank 3 structure from the CSP study? Surely, it is lower energy than rank 4 and seems to be an unobserved structure.

We are sorry for the confusion, which was due to the fact that technically CSP ranks 2 and 3 are the two disordered components of form 1 (see SI Table S3) and rank 4 is ranked 3rd if those two are treated as the same crystal structure rather than as separate predicted polymorphs. We have now edited the manuscript, stating “The lowest energy as-yet-unfound predicted polymorph of ritonavir is ranked third” to make it clearer.

Figure 2: Please state in the caption more information about the meaning of the error bars. The Methods section states “The method outlined by Firaha et al. 8 was finally used to compute the error bar in terms of relative free energy associated with each predicted polymorph of ritonavir.” Are these the standard deviation calculations described in reference 8? If the error bars in the figure are 1 standard deviation, is the statement (page 6) “Note that no predicted polymorph is within the confidence interval of the model of the matches to the known most stable forms” correct? Is this statistically justified based on the observed non-overlap of one standard deviation? Some simple statistical calculations could assess the likelihood, based on these error estimates, that one of the higher energy structures is at or lower in energy than the matches to forms 1 and 2.

This is a very useful comment, thanks for it. We have specified that the error bars correspond to one standard error, as described in reference 8. While it is true that being outside of the error bar of the model is not a full statistical guarantee of instability, the probabilities become quite low. We have also performed the statistical calculations you have suggested. The caption now includes the comment “the error bar, representing one standard error” and the paragraph below has been edited as “Looking at the CSP landscape from a risk assessment perspective, the first noticeable outcome is that no predicted polymorph is within one standard error of the free energy model of the structures matching forms 1 and 2. Statistically, the closest predicted crystal structure has a

chance of about 2.0% and 0.2% of being more stable than forms 1 and 2, respectively.”

p.9. There is a repeated sentence at the top of this page: “This is just the case for single component crystal structures: for solvates, the situation can become even more complicated given the plethora of possible stoichiometries and the possibility of forming non-stoichiometric adducts. 45-47 And this is just the case for single- component crystal structures: for solvates the situation can become even more complicated given the plethora of possible stoichiometries and the possibility of forming non-stoichiometric adducts. 46,47”

We have removed this repetition.

p.9. In discussing solvates and the challenges that they create for CSP, “Although methods have been developed to help with these cases, 48 they require CSP to predict a solvent-free scaffold...” Reference 48 presents a method for hydrates. More general work on solvates was presented in Chemistry–A European Journal 2009, 15 (47), 13033-13040.

While agree, we have removed that sentence in an effort to streamline the discussion as suggested by other reviewers.

p.9. The point about combining CSP with less traditional methods, such as electron diffraction, is important. A important aspect of ED is the small sample size requirement, which means that small amounts of new polymorphs might be able to be identified within the bulk sample of a known form, as shown in Chem. Eur. J., 19: 7883-7888. <https://doi.org/10.1002/chem.201204369>, where a polymorph was identified and structure proposed by comparison to CSP. This has implications in the discussion of risk assessment, and potentially also for intellectual property.

We have emphasized this and added a comment stating: “This ability to characterize crystal structures from a small sample size can also have intellectual property implications”.

Reviewers' comments:

Reviewer #1:

After the major revision, the manuscript clearly demonstrates how CSP can effectively derisk the missing-polymorphic issue in the well-known Ritonavir system, while successfully solving the previously uncharacterized Form 4 structure through CSP and ED techniques. The paper could be recommended for acceptance after addressing follow points.

We thank Reviewer 1 for their comments and recommendation.

P8, it is concluded that Ritonavir is identified as a risk system based on the analysis of CSP result. Is the analysis mentioned in the manuscript based on empirical/workflow-driven subjective judgment, or on qualitative/quantitative computational results? Please provide the analytical details if it is based on qualitative/quantitative computational results in the manuscript or SI.

This analysis is based on a quantitative assessment of the relative free energies of the CSP-generated polymorphs, with the only known crystals structure that was known when ritonavir was being developed, form 1, predicted as less stable at room temperature than the global minimum (which did eventually crystallize as form 2), combined with a quantitative assessment of the potential solubility loss (about 2 fold on average) that could have been caused by the appearance of the more stable form, larger than the formulation could tolerate. To make it clearer, we have modified the sentence in P8 as: "The risk due to CSP predicting a lower energy polymorph than the lead crystalline form 1, as well as the potential consequences of this predicted crystal structure emerging, would have raised significant concerns".

P10, it is noted that the disorder effect of Form 3 was not computed, which limits direct comparability with the disorder analyses performed for Form 1 and 4. It is recommended to supplement this calculation unless there are specific methodological considerations precluding such an analysis.

We did not calculate the disorder effect of form 3 primarily because it would have been very computationally expensive, as form 3 is a $Z'=4$ crystal structure with each of the 4 molecules having disorder and with the disordered groups closely interacting, leading a very large number of potential combinations. Moreover, it may add little benefit, since form 3 is known to be unstable experimentally. Forms 1 and 4 are both $Z'=1$, making the disorder calculation much more realistic, and their relative stability is more relevant, especially as it is experimentally undetermined for form 4. A sentence has been added in P10 mentioning both reasons.

P11/P12, CSP methodology is described. Given its molecular size and conformational flexibility, Ritonavir has traditionally been considered a challenging system for CSP, often regarded as beyond the capabilities of conventional CSP tools/algorithms. Please highlight the key advancements (e.g., methodological improvements, algorithmic optimizations) that enabled the successful prediction of Ritonavir's polymorphic landscape in this study.

This is a very good point, and we agree that ritonavir has been historically viewed as beyond the capabilities of CSP algorithms. We have added a sentence in the Introduction (P3) to explain what

advancements have occurred: “While ritonavir has often been considered beyond the capabilities of the most accurate CSP algorithms, recent developments, including the emergence of faster compute nodes, the increasing availability of Cloud computing, the improvement in force fields allowing a reduction in the number of required quantum-mechanical calculations, and the development of more accurate, computationally feasible methods for free energy calculation at finite temperature, have made it a realistic (albeit challenging) target”, and provided a few references in support.

P12, it is noted that the Fvib was explicitly calculated for 12 crystal structures, while the remaining structures were adjusted using a weighted average of the computed Fvib values. This approach completely neglects the thermal contributions to relative stability among the uncalculated structures and represents an overly simplistic approximation. It is recommended to either explicitly compute Fvib for the remaining crystals or exclude these structures from the 300 K energy landscape and tables.

This is a valid point. However, we believe that including all structures in the landscape is valuable for the reader to see what kind of spread exists in terms of relative stability for the predicted polymorphs of ritonavir; moreover, including higher lattice energy polymorphs in the finite temperature crystal energy landscape by only considering the lattice energy contribution to the relative stability is common in current CSP workflows; see e.g. the group 20 methodology in the SI of “Hunnisett, L. M. et al. The seventh blind test of crystal structure prediction: structure ranking methods. *Acta Crystallogr B Struct Sci Cryst Eng Mater*”. We have referenced this in the manuscript and added an explanation for this decision in P12, stating how this is due to the very large computational cost of the Fvib calculations. In general, most of the relative stability differences are due to lattice energy, with the Fvib contributions being important but only at the margin and therefore unlikely to be decisive for higher lattice energy polymorphs (see Nyman, J. & Day, G. M. Static and lattice vibrational energy differences between polymorphs. *CrystEngComm* **17**, 5154-5165). Finally, we would like to point out that while neglected in the calculations the potential effect of the vibrational contributions on the relative free energy is captured in the larger error bars for the structures where the Fvib was not explicitly calculated.

To enhance understanding of thermal effects on polymorph stability of Ritonavir system, it would be informative to include the relative energies and/or crystal energy landscape at 0 K in SI for comparison.

The 0 K lattice energy landscape has been added in SI Figure S6.

Reviewer #2:

The paper has been revised into a much better version of itself and can be published in this form. We sincerely thank Reviewer 2 for their feedback.

Reviewer #3:

I co-reviewed this manuscript with one of the reviewers who provided the listed reports. This is part of the Communications Chemistry initiative to facilitate training in peer review and to provide appropriate recognition for Early Career Researchers who co-review manuscripts.

We thank Reviewer 3 as well.

Reviewer #4:

I have read the response to reviews and the revised manuscript. My opinion is that the authors have satisfactorily addressed the criticisms from the review of the original manuscript. The revised manuscript is an important contribution to the field, showing that current crystal structure prediction methods are able to address systems like Ritonavir and make predictions that could have averted the problems of late appearing polymorphism. I support publication.

We thank Reviewer 4, and we are grateful for their comment about the importance to the field.

Reviewers' comments:

Reviewer #1:

I have reviewed the authors' response and the revised manuscript. The authors have adequately addressed most of the points. One minor suggestion would be to improve readability and methodological transparency of the revised sentence on Page 13, for example by explicitly stating that "...the finite-temperature F_{vib} was estimated by applying a uniform shift to all values based on the average F_{vib} from the 12 calculated crystals...(see SI Section 4.1 for details)". Should this suggestion be incorporated, I would be pleased to recommend the manuscript for publication.

We thank Reviewer 1 for their feedback and recommendation. We have addressed their suggestion regarding improving readability and methodological transparency in the Methods by editing page 13 consistently with their proposal: "For the remaining less stable predicted polymorphs, given the high computational associated with the phonon frequency calculations, finite temperature F_{vib} was estimated by applying a uniform shift to all lattice energy values based on the average F_{vib} for the 13 for which it had been explicitly calculated".